# NKG2A is a NK cell exhaustion checkpoint for HCV persistence

Chao Zhang[1,2], Xiao-mei Wang[3], Shu-ran Li[1,2,4], Trix Twelkmeyer[1,5], Wei-hong Wang[1,2,4], Sheng-yuan Zhang[1,2,6], Shu-feng Wang[7], Ji-zheng Chen[8], Xia Jin[5], Yu-zhang Wu[7], Xin-wen Chen[8], Sheng-dian Wang[2], Jun-qi Niu[3], Hai-rong Chen[1,2] & Hong Tang[1,5,8]

Exhaustion of cytotoxic effector natural killer (NK) and CD8[+] T cells have important functions in the establishment of persistent viral infections, but how exhaustion is induced during chronic hepatitis C virus (HCV) infection remains poorly defined. Here we show, using the humanized C/O[Tg] mice permissive for persistent HCV infection, that NK and CD8[+] T cells become sequentially exhausted shortly after their transient hepatic infiltration and activation in acute HCV infection. HCV infection upregulates Qa-1 expression in hepatocytes, which ligates NKG2A to induce NK cell exhaustion. Antibodies targeting NKG2A or Qa-1 prevents NK exhaustion and promotes NK-dependent HCV clearance. Moreover, reactivated NK cells provide sufficient IFN-γ that helps rejuvenate polyclonal HCV CD8[+] T cell response and clearance of HCV. Our data thus show that NKG2A serves as a critical checkpoint for HCV-induced NK exhaustion, and that NKG2A blockade sequentially boosts interdependent NK and CD8[+] T cell functions to prevent persistent HCV infection.

[1] The Joint Laboratory of Infection and Immunity at Institut Pasteur of Shanghai and Institute of Biophysics, Chinese Academy of Sciences, Beijing 100101, China. [2] The Key Laboratory of Infection and Immunity, Institute of Biophysics, Chinese Academy of Sciences, 100101 Beijing, China. [3] Department of Hepatology, The First Hospital of Jilin University, 130021 Changchun, Jilin, China. [4] College of Life Sciences, University of Chinese Academy of Sciences, 100049 Beijing, China. [5] The Key Laboratory of Molecular Virology and Immunology, Institut Pasteur of Shanghai, Chinese Academy of Sciences, 200031 Shanghai, China. [6] School of Life Science and Technology, ShanghaiTech University, 200031 Shanghai, China. [7] Institute of Immunology, The Third Military Medical University, 400038 Chongqing, China. [8] The State Key Laboratory of Virology and Center for Viral Pathology, Wuhan Institute of Virology, Chinese Academy of Sciences, 430071 Wuhan, China. These authors contributed equally: Chao Zhang, Xiao-mei Wang. Correspondence and requests for materials should be addressed to H.-r.C. (email: hairong.chen@ibp.ac.cn) or to H.T. (email: htang@ips.ac.cn)

Hepatitis C virus (HCV) infection causes more than 185 million carriers worldwide[1]. During the natural course of HCV infection, spontaneous clearance of the virus occurs in only 15–20% of acutely infected adults, while the remainders develop chronic infection, which often progress to cirrhosis and hepatocellular carcinoma[2]. Exhaustion of HCV-specific CD8+ T cells, characterized by upregulation of co-inhibitory receptors (PD-1, CTLA-4, Tim-3, Lag-3, 2B4, and CD160), may associate with chronic hepatitis C (CHC)[3], with PD-1 being the most studied. However, PD-1 checkpoint inhibitor therapy only induce fairly limited antiviral response in HCV-infected primates (1 of 3)[4] or patients (4 of 20)[5]. In agreement with this, PD-1 blockade in vitro is insufficient to restore the cytotoxicity of hepatic CD8+ T cells isolated from CHC patients[6,7]. Thus, more roadblocks of immune tolerance need to be removed in CHC in addition to PD-1 or cytotoxic CD8+ T lymphocytes (CTL).

Natural killer (NK) cells are an important effector lymphocyte population in anti-tumor and anti-infection immunity[8]. NK cells account for 25–50% of human liver lymphocytes and 5–10% of mouse liver lymphocytes[9], indicating their importance in livers. The activity of NK cells is controlled by an array of activating and inhibitory receptors[10]. A number of studies have highlighted the potential importance of NK cells during HCV infection[11]. In brief, NK cells are activated in the acute phase of HCV infection, with upregulation of the activating receptors (e.g., NKG2D), IFN-γ production and cytotoxicity[12], which associates with the spontaneous clearance of HCV in healthcare workers[13] and intravenous drug users[14]. On the other hand, chronic HCV infection often associates with exhaustion of NK cells, limiting its anti-infection activity. For example, the inhibitory receptor NKG2A is upregulated in the circulating NK cells[15], in line with the reduced IFN-γ production[16] and cytotoxic function[16,17] of intrahepatic NK cells in CHC patients. Another NK inhibitory receptor, KIR2DL3, when present on a homozygous ligand background (HLA-C1/C1) that induces a weaker inhibitory effect easier to be overcome by activation signals, is associated with spontaneous resolution of HCV infection[18]. However, how NK cell exhaustion is induced and maintained early in the infection, and more importantly, whether NK cell exhaustion determines HCV persistence, remain unclear.

By expressing human occludin and CD81 in an outbred ICR strain (C/O^Tg), we have previously generated an immune-competent humanized mouse permissive for HCV persistent infection[19], and have successfully applied to a number of studies[19–23]. Using this mouse model, we show here the dynamics of hepatic infiltration and exhaustion of NK and CD8+ T cells during acute HCV infection. Furthermore, we are able to depict the nature of upregulated hepatic Qa-1 interacting with the inhibitory receptor NKG2A on NK cells to induce NK exhaustion. Anti-Qa-1 or anti-NKG2A antibody treatment restores NK and sequentially CD8+ T cell cytotoxicities in HCV clearance. Our study highlights the importance of Qa-1/NKG2A exhaustion checkpoint, when compared with PD-1/Tim-3, in the establishment of HCV persistence.

## Results

**HCV persistence is associated with CD8+ T cell exhaustion.** Acute HCV infection is characterized by a significant delay in the onset of T cell response[24]. We have shown previously that hepatic infiltrated T cells were generally inactive after HCV infection[19]. Using the same humanized mice model of persistent HCV infection, we repeated the tail vein perfusion of C/O^Tg mice or wt littermates with HCV. Measurement of HCV genome copies in livers indicated the expected progression of acute (1 d–2 w) to persistent (>2 w) infection (Fig. 1a). Luminex measurement of

serum cytokines showed the typical delayed Th1 (IFN-γ, IL-2, and IL-12p40) and an absence of Th2 response (type II cytokines below detection limits) along the course of infection (Supplementary Fig. 1A and B), reminiscent of the observation in patients[25].

We then assessed HCV-specific CD8+ T cell response. ELISpot analysis on splenocytes isolated from wt or C/O^Tg mice showed that HCV multi-specific T cells were elevated preferentially in HCV (genotype 2a) infected C/O^Tg mice, which peaked at 4–7 dpi (see Table S1 for HCV NS3, NS5B, NS5A, and core peptides), and vanished at 2 wpi (Fig. 1b). Striking enough, the decline of HCV-specific T cell response was concomitant with upregulation of PD-1 in both hepatic and peripheral CD8+ T cells (Fig. 1c). To exclude artifacts caused by a particular HCV strain, ELISpot assays were performed after C/O^Tg mice were infected with patient sera positive for HCV1b ($3.93 \times 10^6$ copies/mL). Persistent HCV1b infection in C/O^Tg liver (Supplementary Fig. 2A) correlated with HCV multi-specific T cell exhaustion (Supplementary Fig. 2B; see Table S2 for HCV epitope pools) and PD-1 upregulation in CD4+ and CD8+ T cells, starting at 1 wpi (Supplementary Fig. 2C). Of note, PD-1 upregulation was faster and more pronounced in hepatic CD8+ T cells than peripheral CD8+ T cells (Fig. 1c and Supplementary Fig. 2C). Taken together, these results suggest that CD8+ T cell exhaustion concurred with HCV persistent infection in C/O^Tg mice, reminiscent to CHC patients[26].

Previous studies with PBMC of CHC patients showed PD-L1/PD-1 upregulation associated with HCV-specific T cell exhaustion[27]. We further showed that PD-L1 was increased in C/O^Tg hepatocytes 2 wpi (Fig. 1d), presumably by the transient activation of type I or II IFNs upon HCV invasion[28]. To substantiate whether PD-L1/PD-1 signaling was critically involved in establishment and/or maintenance of CD8+ T cell exhaustion, PD-1 blocking antibody was i.p. administrated to C/O^Tg mice 1 day before HCV inoculation and continued for 2 or 4 weeks (Fig. 1e). The results indicated that PD-1 intervention slightly reduced viral loads in the serum but not liver at 2 wpi, and a prolonged treatment had no benefit (Fig. 1f). Failure to clear HCV by PD-1 inhibitor was associated with negligible change of HCV-specific T cell activities (Fig. 1g). Recent in vitro studies suggest that a combined blockade of PD-1 and CTLA-4[6] or Tim-3[29] can reactivate HCV-specific CD8+ T cells. Tim-3 was also upregulated in intrahepatic CD8+ T cells of HCV-infected C/O^Tg mice (Fig. 1h). However, a dual administration of blocking antibodies to PD-1 and Tim-3 failed to reduce HCV loads in C/O^Tg mice (Fig. 1i). Therefore, PD-1/Tim-3 blockade was insufficient to reverse CD8+ T cell exhaustion, or alternatively, more blocks of immune tolerance, dependent or independent of T cell exhaustion in livers, needs to be overcome to eliminate HCV.

**Impaired hepatic NK cell function leads to HCV persistence.** Restricted by availability and accessibility to liver tissues of patients with acute infection, the role of intrahepatic NK cells in the establishment of HCV persistence remains largely speculative. We previously showed that NK cells, but not NKT or other myeloid cells, were recruited to the liver 12 h post HCV infection in C/O^Tg mice[19]. Interestingly, the acute liver infiltration of NK cells coincided perfectly with the sharp elevation of chemokines known for NK cell chemotaxis (MCP-1/CCL2, KC/CXCL1, MIG/CXCL9, and IP-10/ CXCL10) within 2 dpi (Supplementary Fig. 1C). For hepatic non-parenchymal cells (NPC) isolated from the same set of mice in Fig. 1a, we examined the phenotypic profile of NK cell functions. Only in infected C/O^Tg but not abortively infected wt mice, circulating cytokines responsible for NK differentiation (IL-12p40; Supplementary Fig. 1A) and NK

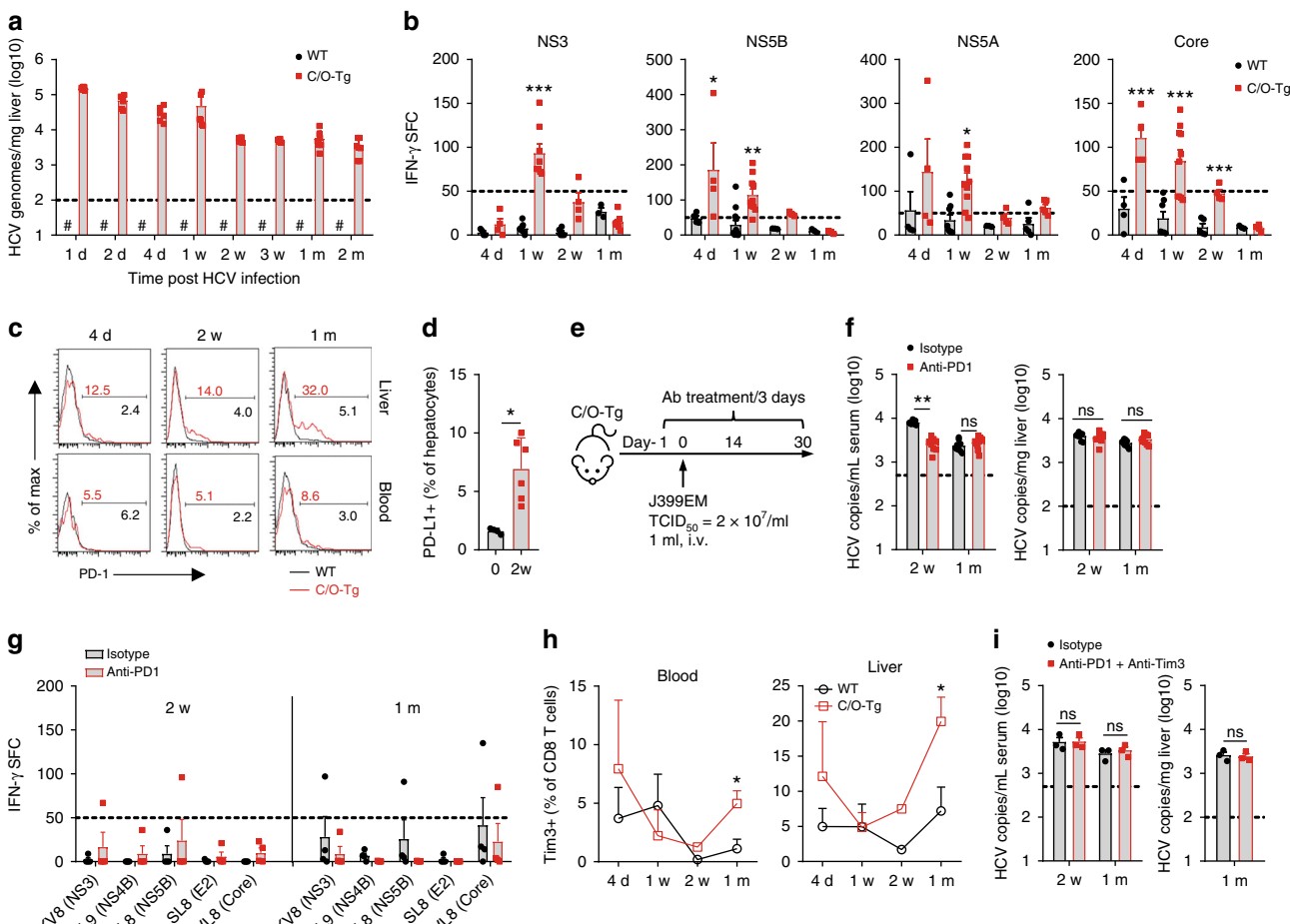

**Fig. 1** CD8[+] T cell exhaustion and PD-1 blockade. C/O[Tg] ($n = 55$, $n \geq 6$ for each time point) or wt mice ($n = 32$, $n = 4$ for each time point) were i.v. infused with 1 mL HCV J399EM (TCID$_{50}$ = $2 \times 10^{7}$/mL). Mice were killed and blood, splenocytes, hepatocytes, and NPCs were collected at indicated time. **a** HCV RNA copies in hepatocytes were measured by qPCR. **b** The number of IFN-γ spot-forming cells (SFCs) by ELISpot assay determined 2.5 d after in vitro stimulation of splenocytes with respective epitope peptides, subtracted the SFC of OVA stimulation as the background. **c** FACS measurement of PD-1 expression in circulating or hepatic CD8[+] T cells at indicated time post HCV infection. **d** FACS analysis of PD-L1 expression on hepatocytes isolated from naive ($n = 4$) or HCV-infected C/O[Tg] mice ($n = 6$) 2 wpi. **e** C/O[Tg] mice ($n = 9$ for each group) were i.p. injected with PD-1 blocking antibody or isotype Ig (200 μg/3 days) 1 day before HCV inoculation. **f** HCV RNA copies in serum and livers, and **g** HCV-specific T cells response to the indicated epitopes. **h** FACS analysis of Tim-3 expression on peripheral and liver CD8[+] T cells. **i** qPCR measurement of HCV RNA copies in the liver or blood after C/O[Tg] mice ($n = 3$ for each group) were i.p. injected with PD-1 (200 μg/3 days) plus Tim-3 (100 μg/2 days) blocking antibody or isotype IgG 1 day before HCV inoculation. Dash lines indicated limits of detection of related assays (qPCR: 100 copies/mg liver, 500 copies/mL serum; ELISpot: 50 spots/4 × 10[5] splenocytes). #, below detection limit. Data were mean ± SD, Student t-test in **b**, **d**, **f**, **h**, and **i**. *$P < 0.05$; **$P < 0.01$; ***$P < 0.001$. ns Not significant. Source data are provided as a Source Data file

function (IFN-γ, RANTES/CCL5; Supplementary Fig. 1B) peaked in the first two days and quickly declined to the baseline within 4 dpi. FACS analysis of NPC confirmed an increase of the conventional DX5[+] NK subset (Fig. 2a) and homologous functional CD11b[+] population (Fig. 2b) within 4 dpi (see Supplementary Fig. 3 for gating strategy), which concurred well with upregulation of NK activating receptors, Ly49D, Ly49H, and NKG2D, in C/O[Tg] mice (Fig. 2c). These results suggest a transient hepatic infiltration and activation of NK cells in response to HCV infection. More importantly, these activating receptors descended at 4 dpi, which yielded to the increase of the inhibitory receptors, KLRG1, NKG2A, and TIGIT in hepatic NK cells, and sustained to 2 mpi (Fig. 2d). Several other inhibitory markers, including Ly49 receptors and Tim-3, were not responsive to HCV infection (Supplementary Fig. 4). HCV upregulation of NKG2A and KLRG1 was also observed for peripheral NK cells (Fig. 2e). Intrahepatic NK cell exhaustion along the course of HCV infection was then confirmed by NK function assay in vitro. Results

showed that intracellular IFN-γ and CD107a of NK cells isolated from C/O[Tg] livers elevated within 4 dpi, which rapidly declined to the baseline as the un-infected hepatic NK cells afterwards (Fig. 2f). In parallel, CD107a and granzyme B were also reduced in NK cells when C/O[Tg] mice progressed to persistent HCV1b infection (Supplementary Fig. 2D). Therefore, liver-infiltrated NK cell exhaustion might attribute to HCV persistent infection in C/O[Tg] mice.

To delineate which inhibitory receptor(s) accounted for NK cell exhaustion in the liver, we analyzed carefully the cohort of C/O[Tg] mice with different outcomes of HCV infection (Fig. 3a, b). Mice with spontaneous clearance of HCV exhibited a very low NKG2A expression in peripheral NK cells, whereas 40% of mice with persistent HCV infection showed a much higher NKG2A level (Fig. 2c). Other receptors (NKG2D and TRAIL; Supplementary Fig. 5A) or functional cytokines (G-CSF, IP-10, MIG, CXCL5; Supplementary Fig. 5B) of NK cells were not differently expressed. PD-1, KLRG1, and Tim-3 receptors in peripheral

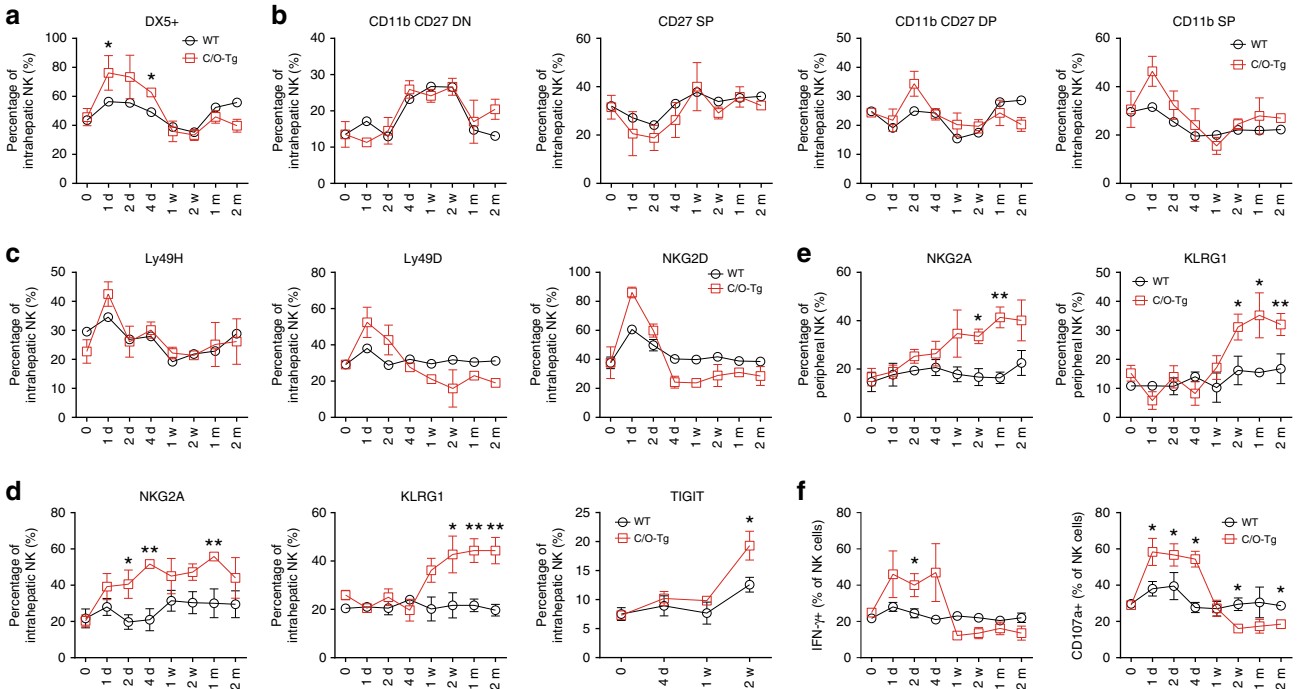

**Fig. 2** Impaired hepatic NK cell response during HCV persistent infection in C/O[Tg] mice. Mice were treated as in Fig. 1a. FACS analyses of intrahepatic NK cells for expression of **a** DX5, **b** CD11b, and CD27, **c** activation receptors, including Ly49D, Ly49H, and NKG2D, and **d** inhibitory receptors, including NKG2A, KLRG1, and TIGIT. **e** FACS analyses of peripheral NK cells for expression of NKG2A and KLRG1. **f** NK function assays were carried out at indicated time of HCV infection by Yac-1 cell stimulation. CD107a and IFN-γ expression were FACS analyzed. Data were mean ± SD, ANOVA test. *P < 0.05; **P < 0.01. Source data are provided as a Source Data file

CD8[+] T cells failed to differentiate self-limiting from persistent infection, either (Supplementary Fig. 5C). Therefore, NKG2A upregulation, potentially impaired NK function, might contribute to the establishment of HCV persistence.

To determine whether NKG2A signaling was required for HCV persistence, a blocking antibody against NKG2A was i.p. administrated 1 day before HCV infection and continued once every 3 days until 1 or 2 wpi when C/O[Tg] mice were killed (Fig. 3d). NKG2A blockade resulted in a significant reduction of HCV replication in livers and the serum at 1 and 2 wpi (Fig. 3e). The reduced HCV replication correlated with the increased NK degranulation (Fig. 3f) and NK killing activities (Fig. 3g) after anti-NKG2A treatment. Therefore, we suspected that NKG2A blockade would benefit a robust clearance of HCV infection from preventing or reversing of NK exhaustion in the liver. To further test whether NKG2A blockade was effective to impede the progression of acute to persistent HCV infection, anti-NKG2A was employed 2 wpi, when the elevated expression of NKG2A had already been observed (Fig. 2d, e), and continued for additional 2 weeks (Fig. 3h). NKG2A inhibition reduced HCV levels in sera and livers (Fig. 3i), accompanied by not only an elevated hepatic NK cell activity (Fig. 3j), but also HCV-specific T cells response (Fig. 3k). Therefore, in HCV chronic infection, targeting NKG2A would break the immune tolerance of both NK and HCV-specific T cell response. Of note, anti-NKG2A antibody (clone 20D5) also recognizes NKG2C and NKG2E, two activating receptors of NKG2 family members[30]. However, the effect of the antibody to reverse NK cell exhaustion suggests that NKG2A checkpoint inhibition was the major mechanism involved.

**NK and T cells interplay contributes to HCV persistence**. The previous[14,31] and aforementioned results suggest that an increased NK function positively correlated with the magnitude of virus-specific T cell responses, and exhaustion of NK and

T cells would both associate with HCV persistence. It is therefore critical to determine the nature of NK and T cells interplay in the course of HCV infection. Intriguingly, inactivation of NKG2A signaling readily reversed NK cell exhaustion (Fig. 3g) and reduced HCV viral loads within 1 wpi (Fig. 3e), when HCV multi-specific T cell responses were not fully restored (Fig. 4a). On the other hand, PD-1 blockade failed to restore NK cell function (Supplementary Fig. 6). Therefore, impairment of NK cell function would precede T cells exhaustion in the establishment of HCV persistent infection. NKG2A is also expressed in T cells. To substantiate that NKG2A upregulation in NK cells was specifically required for signaling to T cell exhaustion, we depleted NK cells (Supplementary Fig. 7A for depletion efficacy) prior to anti-NKG2A treatment in C/O[Tg] mice. The results showed that, in the absence of NK cells, NKG2A blockade failed to restore HCV-specific T cell activities (Fig. 4b), and with a pronounced elevation of HCV viral loads (Fig. 4c). Restoration of HCV-specific T cells cytotoxicity by anti-NKG2A would thus depend on NK cells. On the other hand, elimination of HCV by reactivated NK cells after anti-NKG2A treatment requires CD8[+] T cells. This is because a prior depletion of CD8[+] T cells in C/O[Tg] mice (Supplementary Fig. 7B for depletion efficacy) abolished the effect of anti-NKG2A on HCV replication inhibition (Fig. 4c). Therefore, impairment of NK cell function would lead to CD8[+] T cell exhaustion in the establishment of HCV persistent infection.

**HCV upregulated hepatic Qa-1 impairs NK function**. HCV virus per se does not affect NK function[32], rather that HCV-infected hepatocytes induce functional activation[33] or impairment of human NK cells[34,35]. These results probably stem from in vitro co-culture of NK cells and HCV-infected cell lines, which are MHC-mismatched, transformed or natural immunogenic to NK cells. To validate the interaction between HCV-infected liver cells and NK cells, we devised a NK function assay by co-culturing the

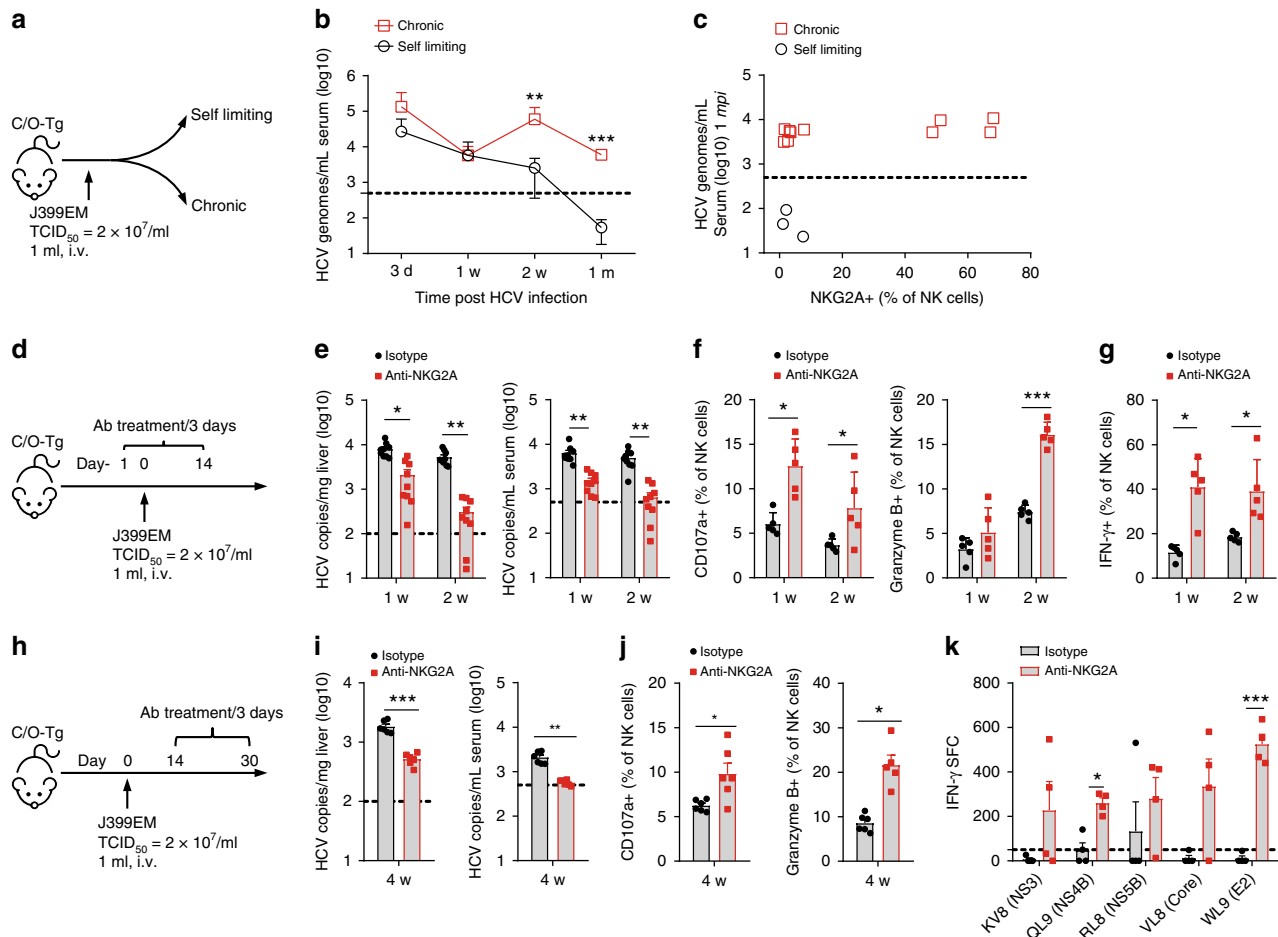

**Fig. 3** NKG2A blocking promoted HCV clearance. **a** C/O$^{Tg}$ mice ($n = 13$) were infected with 1 mL HCV J399EM (TCID$_{50}$ = 2 × 10$^7$/mL). **b** The self-limited infection in C/O$^{Tg}$ mice exhibited an initial high peripheral viral load (<1 mpi) followed with RNA copies below the detection limit (≥1 mpi), whereas HCV persistent infection was characterized by sustained HCV viral loads in the serum beyond 1 mpi. **c** FACS analysis of NKG2A in peripheral NK cells and qPCR measurement of HCV RNA copies in the serum 1 m after C/O$^{Tg}$ mice ($n = 13$) were infected. **d** C/O$^{Tg}$ mice ($n = 9$ for each time point) were treated with anti-NKG2A or isotype Ig (100 μg/3 days, i.p.) 1 day before HCV infection. **e** qPCR measurement of HCV RNA copies in livers and sera; **f** FACS analysis of CD107a and granzyme B expression in NK cells; **g** FACS analysis of intracellular IFN-γ in isolated NK cells after co-cultured with Yac-1. **h** C/O$^{Tg}$ mice ($n = 6$ for each time point) were treated with anti-NKG2A or isotype Ig (100 μg/3 days, i.p.) beginning at 2 wpi, **i** HCV RNA copies in livers and sera; **j** CD107a and granzyme B were measured in NK cells; and **k** HCV-specific CD8$^+$ T cell response by IFN-γ ELISpot assays using splenocytes (4 × 10$^5$ for each well) isolated from mice 1 mpi. Epitope peptides (final concentration 4 μg/mL) were indicated. Dash lines indicated limits of detection (qPCR, 500 copies/mL serum or 100 copies/mg liver; ELISpot, 50 spots/4 × 10$^5$ splenocytes). Data were mean ± SD, Student *t*-test. *$P < 0.05$; **$P < 0.01$; ***$P < 0.001$. Source data are provided as a Source Data file

isolated NPC or purified NK cells with the matched HCV-infected primary hepatocytes (PHT) (Fig. 5a). C/O$^{Tg}$ PHT were fully supportive of HCV infection (Supplementary Fig. 8A and B), as previously described[19]. The addition of HCV-infected PHT, but not naive PHT, downregulated expression of CD107a and secretion of IFN-γ by NK cells, which had been stimulated by Yac-1 target cells or PMA and ionomycin (P + I) treatment (Fig. 5b). Therefore, HCV-infected hepatocytes impair NK cell function.

How HCV-infected PHT cells impaired NK cell function remains unknown. We previously observed a transient activation of type I/III IFNs and IL-10 in acute HCV infection of C/O$^{Tg}$ mice[19]. Upregulated PD-L1 and IL-10 in DC and macrophages by type I interferons (IFN) may contribute to T cell exhaustion in the establishment of LCMV viral persistence[36,37]. IL-10 is also shown to upregulate NKG2A in NK cells in chronic HBV infection[38]. To investigate whether these cytokines may modulate NK cell function in response to HCV infection, we repeated the assay as in Fig. 5a, in presence of various blocking antibodies as indicated (Fig. 5c). Surprisingly, blockage of NKG2A (anti-NKG2A) and Qa-1, the ligand of NKG2A in mice, effectively alleviated the functional impairment of NK cells induced by HCV-infected PHT, when compared with isotype antibody control (Fig. 5c). In contrast, antibody blocking of type I/III interferon (anti-IFNαR1 and anti-IL28) or IL-10 (anti-IL10R) signaling failed to do so (Fig. 5c). These results suggest that Qa-1/NKG2A ligation would be directly involved in functional impairment of hepatic NK cells after HCV infection. As a control, addition of the supernatant of HCV-infected PHT (CM−PHT + HCV) (Fig. 5d) or transwell separation of HCV-infected PHT from NPC (Fig. 5e) no longer inhibited NK killing activities. Therefore, HCV-infected hepatocytes inhibit NK cell function via cell–cell contact manner, most likely through Qa-1/NKG2A interaction. Other NPC cells, such as monocytes, DC, B, or T cells, were ruled out for their involvement in modulating NK activities by using purified NK cells instead of NPC in the co-culture system (Fig. 5e). These results thus indicate that HCV infection of hepatocytes impairs hepatic NK cells function likely through Qa-1/NKG2A ligation.

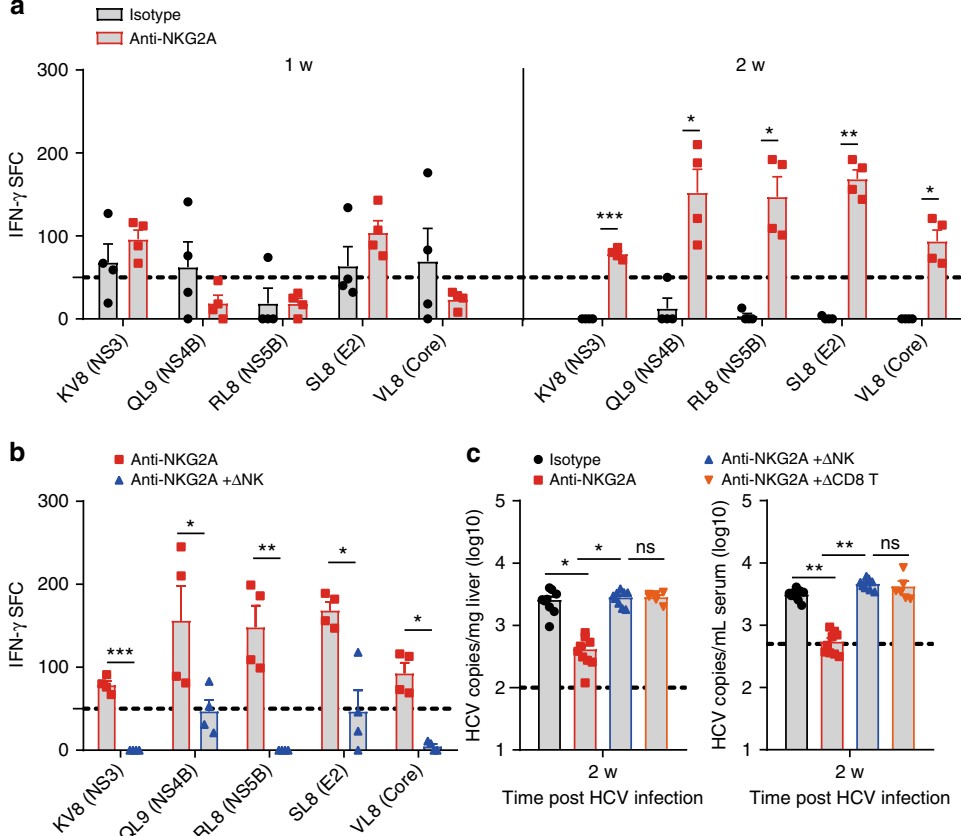

**Fig. 4** NK cells instructed T cells in HCV persistence. **a** Mice were treated as in Fig. 3d. ELISpot assays using splenocytes and the indicated epitope peptides. C/O$^{Tg}$ mice were i.p. pre-treated with mock IgG ($n = 9$) or **b** anti-AGM1 (20 μg/3 days, $n = 9$), **c** anti-CD8 (100 μg/3 days, $n = 6$), together with anti-NKG2A antibody 1 d before HCV inoculation. **b** HCV-specific CD8$^+$ T cell response by IFN-γ ELISpot and **c** hepatic and serum HCV RNA copies were measured 2 wpi. Dash lines indicated limits of detection of related assays (qPCR, 500 copies/mL serum or 100 copies/mg liver; ELISpot, 50 spots/4 × 10$^5$ splenocytes). Data were mean ± SD, Student t-test. *$P < 0.05$; **$P < 0.01$; ***$P < 0.001$. ns Not significant. Source data are provided as a Source Data file

Human HLA-E or mouse Qa-1 interacts with NKG2A to limit NK function. Indeed, HLA-E is upregulated in hepatocytes of CHC patients[39]. In line with this, mRNA level of Qa-1 was also rapidly increased in the liver of HCV-infected C/O$^{Tg}$ mice (Fig. 6a) or HCV-infected PHT (Supplementary Fig. 8C). Flow cytometry showed that Qa-1 upregulation was mainly restricted to hepatocytes but not intrahepatic immune cells (Fig. 6b). Furthermore, administration of a blocking antibody to Qa-1 (Fig. 6c) inhibited HCV replication in C/O$^{Tg}$ mice (Fig. 6d), which simultaneously restored NK functions (Fig. 6e) and cytotoxic CD8$^+$ T cell response, particularly to QL9 epitopes (Fig. 6f). To better characterize the role of Qa-1 in vivo, we delivered cholesterol conjugated siRNA by tail vein injection[40] (Fig. 6g), which selectively downregulated Qa-1 in hepatocytes but not NPC (Supplementary Fig. 9). The partially reduced Qa-1 expression on hepatocytes sufficiently inhibited HCV replication (Fig. 6h), likely by activating NK cell function (Fig. 6i). Taken together, these results indicate that Qa-1/NKG2A interaction served as an innate immune checkpoint that impaired NK function, leading to subsequent T cell exhaustion, in establishment of HCV persistence.

**Revived NK cells alleviate CD8$^+$ T cell exhaustion by IFN-γ.**
The aforementioned evidence suggests that the instructive role of NK cells on T cells, via cytolysis activity or cytokine secretion[41], would have a role in the establishment of HCV persistence. To determine how NK cells regulated HCV-specific T cell response, we first tested whether the direct NK lysis of HCV-infected

hepatocytes was involved in acute HCV infection, which is known to promote myeloid DC (mDC) maturation and antigen presenting capability, and therefore T cells activation[42]. However, liver infiltration (Supplementary Fig. 10A), maturation (CD40, CD80, CD86, and ICOSL, Supplementary Fig. 10B) or activation state (IL-12p40, Supplementary Fig. 10C) of mDC remained invariable, regardless of NK activities impaired or restored by anti-NKG2A treatment. Therefore, mDC cells would unlikely mediate NK-T signaling in HCV infection. Because anti-NKG2A revived IFN-γ secreting capability of NK cells (Fig. 3g), we expected that NK might mediate activation of T cell response through IFN-γ. Intracellular staining indicated that the majority of IFN-γ producing NPCs were NK, NKT, and T cells, but only NK cells showed a pronounced increase of IFN-γ upon anti-NKG2A treatment (Fig. 7a). On the other hand, antibody blockade of IFN-γ completely ablated the capacity of anti-NKG2A in revigorating HCV-specific T cell response (Fig. 7b) and reducing HCV replication (Fig. 7c). Therefore, these results suggest IFN-γ secretion by NK cells would prevent the functional exhaustion of HCV-specific CD8$^+$ T cells, with DC cell dispensable. Because both activated NK and CD8$^+$ T cells were required to clear HCV infection, the reduced HCV viral loads (Fig. 7c) would require IFN-γ from both cells.

## Discussion
Receiving both portal vein and arterial blood, the liver is an important and critical component in the defense against blood-borne infections. The liver acquires specialized mechanisms of

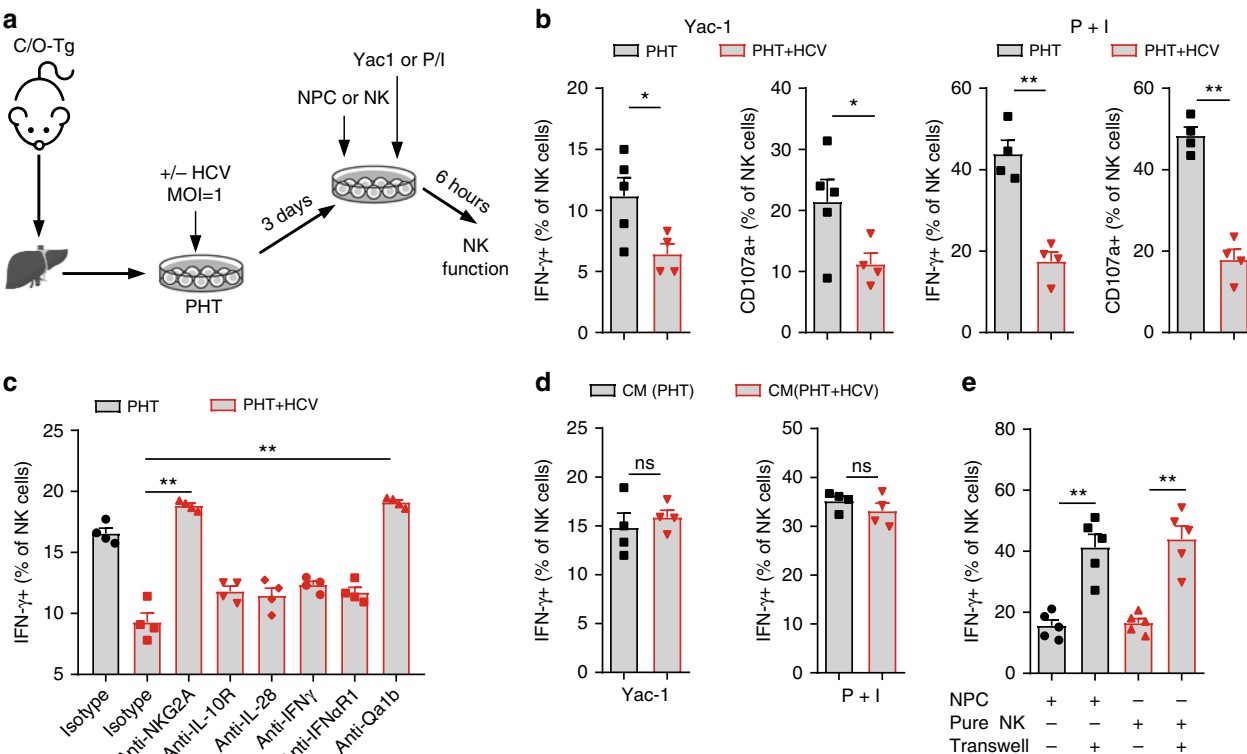

**Fig. 5** HCV infection of hepatocytes impaired hepatic NK cells function. **a** Flowchart of NK functional inhibition assays. PHTs ($2 \times 10^5$/mL) isolated from naive C/O[Tg] mice were infected with HCVcc (MOI = 1) for 3 days before NPCs ($1 \times 10^6$) were added. Yac-1 cells ($1 \times 10^5$/well) or PMA (50 ng/mL) plus ionomycin (1 μM) were used to stimulate NK cells. **b** FACS analysis of intracellular IFN-γ and CD107a of NK cells 6 h after stimulation. **c** Indicated blocking antibodies and isotype IgG were added to the PHT/NK co-culture assay. IFN-γ expression of NK cells was detected 6 h after Yac-1 cell stimulation. **d** Measurement of intracellular IFN-γ in NK cells after supernatants of PHTs after HCV infection (HCV-CM) were added to NPCs. Yac-1 cells ($1 \times 10^5$/well) or PMA (50 ng/mL) plus ionomycin (1 μM) were used to stimulate NK cells. **e** NPCs or MACS purified intrahepatic NK cells were co-cultured with or transwell separated from HCV-infected PHTs. IFN-γ expression of NK cells was detected 6 h after Yac-1 cell stimulation. Data were mean ± SD, Student *t*-test. *$P < 0.05$; **$P < 0.01$. ns Not significant. Source data are provided as a Source Data file

immune tolerance[43], broadly by insufficient antigen presentation and a network of active immunosuppressive pathways mediated mostly by myeloid cells. Detailed mechanisms of how liver tropic viruses, such as HBV and HCV, evade the immune surveillance to establish persistent infections remain vague[44]. Most studies rely on phenotypical or functional assays of PBMC or limited liver biopsies of CHC patients. Inadequate type I IFN, impaired DC presenting, impaired activation of NK and HCV-specific T cells have all been implicated to associate with HCV chronicity[45]. Only scarce clinical evidence suggests a correlation between peripheral NK activation and self-limited HCV infection[13,14], but the dynamic function of intrahepatic NK cells has not been carefully studied. Furthermore, the ineffective virological response to IFN therapy suggests a more profound immune suppression needs to be overcome in CHC patients[46].

Our study reveals that the sustained upregulation of NKG2A in intrahepatic NK cells had a major role in the induction and maintenance of NK cell exhaustion, which eventually leads to HCV persistence. Qa-1/NKG2A-mediated hepatocyte–NK cell interaction provided a insight of how HCV triggered tolerant milieu in the onset of HCV persistence. Plate-immobilized recombinant HCV E2 protein or HCV virus can inhibit cytokines-activated NK functions in vitro, but HCV in a live virion configuration is unlikely to suppress NK cells activity[32,34,47,48]. Instead, this work showed, in support of previous notion[49], that the physical interaction between NK cells and hepatocytes might account for NK cell exhaustion. HCV-infected hepatocytes can avoid NK recognition and cytotoxicity by

upregulation of MHC I and release of specific inflammatory milieu[50]. We identify in this work that NK cells were functionally exhausted by the increased NKG2A/Qa-1 interaction in a cell–cell contact fashion. Abrogation of either NKG2A or Qa-1 signaling, therefore, was sufficient to prevent NK cell exhaustion and inhibit HCV replication (Fig. 7d). Aberrant NKG2A/Qa-1 signaling for NK cell exhaustion is also implicated in other chronic viral infections, where MCMV[51] or HSV-1[52] may escape immune surveillance through upregulation of Qa-1. Elevated HLA-E/NKG2A-mediated inhibition impairs NK cell clearance of HIV-infected target cells[53]. Of relevance, Treg cell-derived IL-10 can elevate NKG2A in NK cells in a mouse model of chronic HBV infection, and NKG2A blockade limits HBV replication[38]. Taken together, NKG2A may serve as an important innate immune checkpoint in establishment of chronic viral infections.

Both NK and T cells express NKG2A, and recent studies indicate that NKG2A checkpoint inhibition also promotes anti-tumor function of both NK and T cells[54,55]. We establish that NK cell activation precedes T cell response to HCV, and reversal of NK cell exhaustion helped reactivating HCV multi-specific CD8[+] T cell response. Therefore, NK cell exhaustion would predominate the tolerant milieu in HCV persistent infection. NKG2A in intrahepatic NK cells, but not T cells, would account for the establishment of HCV persistent infection. This role of intrahepatic NK cells may be unique to HCV, because other studies show that activated NK cells can directly lyse LCMV-specific CD4[+] T cells, leading to CD8[+] T cell exhaustion and LCMV persistence[56,57]. Moreover, nucleos(t)ide analog (NUC)

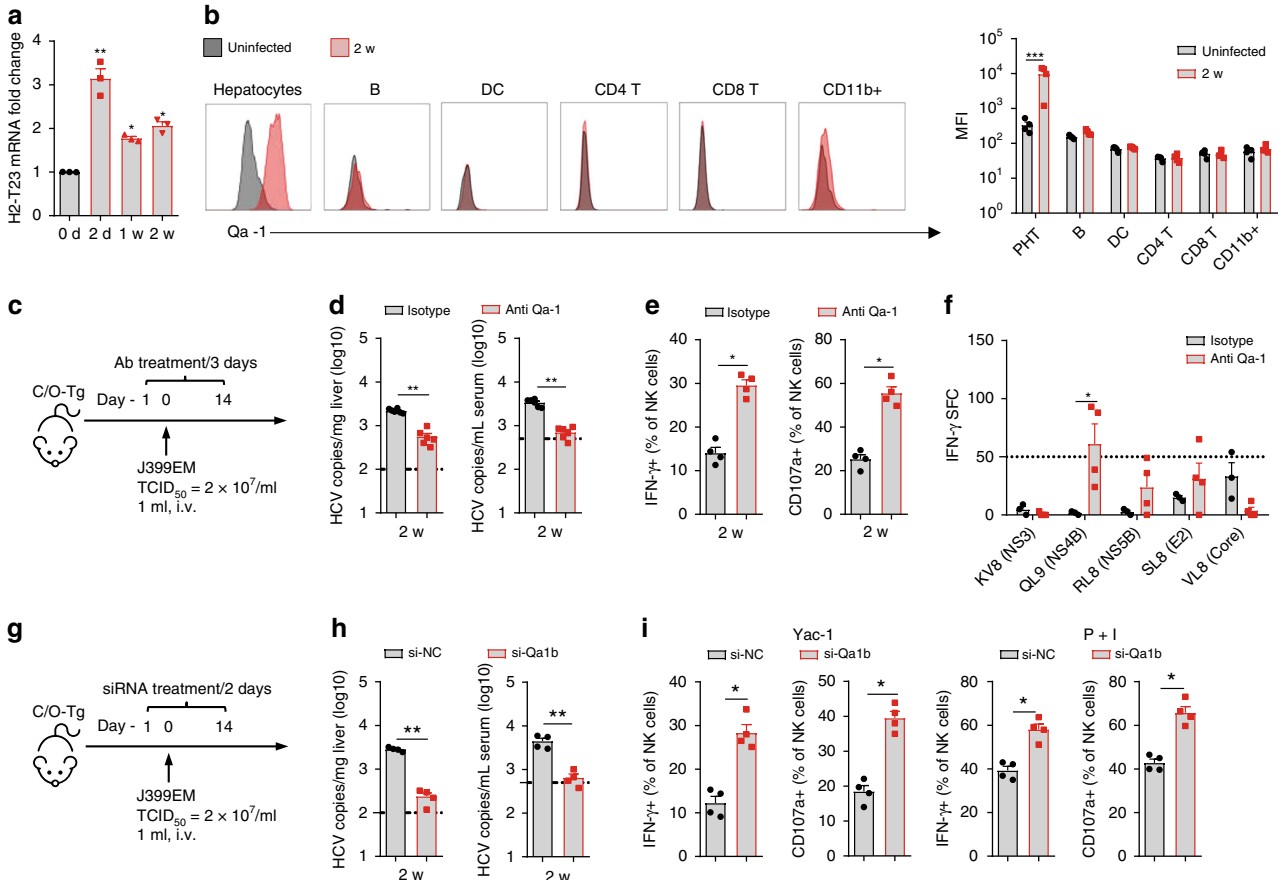

**Fig. 6** Upregulation of Qa-1 on hepatocytes accounted for NK cell dysfunction upon HCV infection. **a** Measurement of Qa-1 mRNA in liver isolated from C/O[Tg] mice (*n* = 3) at the indicated time post HCV infection. **b** FACS analysis of Qa-1 expression on hepatocytes or NPC subsets as indicated before or 2 weeks post HCV infection. **c** C/O[Tg] mice (*n* = 6 each group) were i.p with Qa-1 blocking antibody or isotype IgG (100 µg/3 days). **d** HCV RNA copies in serum and liver, **e** CD107a and IFN-γ expression in hepatic NK cells upon Yac-1 cell stimulation, **f** IFN-γ ELISpot assays of T cells stimulated by the indicated epitopes, were assayed 2 wpi. **g** C/O[Tg] mice (*n* = 4 each group) were i.v. injected with liposomes/siRNA complex at a dose of 2.5 mg/kg every 3 days since day −1. Mice were killed at 2 wpi. **h** HCV RNA copies in serum and liver, **i** CD107a and IFN-γ expression in hepatic NK cells upon Yac-1 cell or P + I stimulation as indicated. Dash lines indicated limits of detection of related assays. Data were mean ± SD, Student *t*-test. *P < 0.05; **P < 0.01; ***P < 0.001. ns Not significant. Source data are provided as a Source Data file

therapy restores HBV-specific T cells function[58] with concomitant inactivation of NK functions[59]. NK depletion, as well as TRAIL and NKG2D blockade, can significantly enhance HBV-specific T-cell functions[59]. Further, NK lytic activity on T cells may be counteracted by Qa-1 in CD4+ T cells signaling to NKG2A[60]. Therefore, the action of Qa-1/NKG2A in chronic HCV infection of C/O[Tg] mice revealed a different mechanism, whereas NKG2A blockade helped to prevent NK cell exhaustion and reverse CD8+ T cell exhaustion, likely through increased secretion of IFN-γ by NK cells. Such a "one stone and two birds" effect of NK checkpoint therapy was recently suggested in TIGIT blockage therapy, which simultaneously prevents NK cell exhaustion and restores CD8+ T cell-mediated anti-tumor immunity[61], albeit NK-T interaction mechanism has not been described. Therefore, the unique nature of NK cell activation and exhaustion, and the distinct cytokine milieu may determine different outcomes of NK-T interaction in different sets of persistent infections[41].

In sum, we were able to take full advantage of the immune-competent mouse model of HCV natural infection, to decipher the dynamic NK-T cell interactions within the liver along the course of infection. Qa-1/NKG2A interaction linked the pathogen to host immunity, and accounted for impaired NK/T functions in the establishment of HCV persistence. Direct-acting antiviral medications (DAA) are becoming a promising cure of hepatitis C. However, emerging data indicate that individuals who have been cured with DAAs remain susceptible to reinfection[62,63]. This may be largely caused by a limited restoration of the protective immunity, as indicated in a study with chimpanzee model of HCV infection[64]. NKG2A checkpoint inhibitor treatment can boost both innate and adaptive immune responses. Considering that NKG2A antibody (monalizumab) has already been in numerous clinical trials on treatment of rheumatoid arthritis, cancer and stem-cell transplantation[65], a combined NKG2A checkpoint treatment and DAA therapy would avoid the risk of late relapse or reinfection after achieving a sustained virological response via DAA therapies.

## Methods
**Virus stocks**. HCVcc stocks were prepared from a J399EM infectious clone[66] and virus titers (TCID$_{50}$) in supernatants and cell lysates were measured by endpoint dilution assays (EPDA)[67]. Sera positive for HCV1b were obtained from CHC patients before IFN/ribavirin therapy. Individuals with a history of HBV, HEV, HDV, HIV, recent infectious diseases, or other inflammatory diseases (such as rheumatoid arthritis, diabetes, autoimmune hepatitis, hypertension, kidney disease) were excluded. Written informed consent was obtained from all patients, and the experimental protocol was approved by the Ethics Committee of the First Hospital of Jilin University, China (approval code: 130801-067).

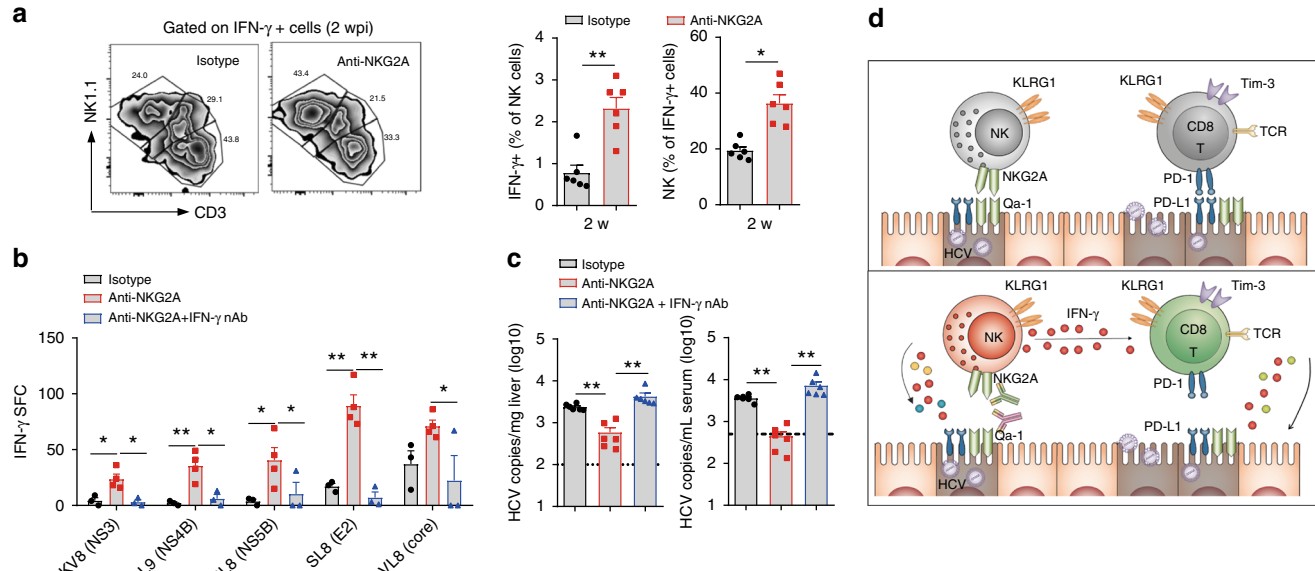

**Fig. 7** Anti-NKG2A restored T cell response through enhanced IFN-γ production. **a** FACS analysis of hepatic IFN-γ positive cells with or without NKG2A blockade 2 wpi. **b**, **c** C/O$^{Tg}$ mice ($n = 6$ each group) were i.p. pre-treated with IFN-γ neutralization antibody or isotype IgG (100 μg/3 days) together with anti-NKG2A antibody 1 d before HCV inoculation. **b** HCV-specific CD8$^+$ T cell response by IFN-γ ELISpot and **c** serum and hepatic HCV RNA copies were measured 2 wpi. **d** Diagram of Qa-1/NKG2A immune checkpoint during HCV infection. Dash lines indicated limits of detection of related assays. Data were mean ± SD, Student $t$-test. $*P < 0.05$; $**P < 0.01$. Source data are provided as a Source Data file

**Mice**. Immune-competent C/O$^{Tg}$ mice that support HCV persistent infection with appropriate hepatopathogenesis were used in this study[19]. For in vivo experiments, 8- to 12-week old, age- and gender-matched C/O$^{Tg}$ and wt littermates were used. Wt mice were littermates by crossing founder C$^{Tg}$ and O$^{Tg}$ mice[19]. All mice were maintained under specific pathogen-free conditions and treated in accordance to protocols approved by the Institute of Biophysics, Chinese Academy of Sciences Institutional Laboratory Animal Care and Use Committee. Infection experiments were conducted in a BSL2-enhanced Animal Care facility. Hereby mice were tail vein perfused with HCV J399EM or HCV patient sera (Genotype 1b) as described previously[19,21]. There was no blinding in infection experiments. Genders of mice used in different assays were random. For experiments that HCV-specific CD8$^+$ T cell were examined by ELISpot, C/O$^{Tg}$ mice positive for similar levels of H-2 class I histocompatibility antigens (H-2-K$^b$, H-2-K$^d$, H-2-K$^k$) and NKG2A were used.

**Cell depletion and inhibition of surface proteins in vivo**. C/O$^{Tg}$ mice were i.p. treated with anti-NKG2A/C/E (100 μg, q.3d. clone 20d5; eBioscience) or anti-Qa-1 (100 μg, q.3d. 6A8.6F10.1A6; BD Pharmingen) to block the NKG2A–Qa-1 interaction. Where indicated, NK depletion antibody (20 μg, q.3d. clone Poly21460, BioLegend), CD8$^+$ T depletion antibody (100 μg, q.3d. clone TIB210) or anti-IFN-γ antibody (100 μg, q.3d. clone R4-6A2, eBioscience) was i.p. injected 1 day before C/O$^{Tg}$ mice or wt littermates were infected with HCV. PD-1 antibody (200 μg, q.3d. clone G4, kindly provided by Prof. S. Wang, Institute of Biophysics, CAS) and Tim-3 antibody (100 μg, q.2d. clone BE0115, BioXcell) were i.p. administrated alone or in combination as indicated 1 day before HCV inoculation, for 2 weeks or 1 month. For siRNA knockdown in vivo, on three consecutive days, Qa-1 or negative control (NC) cholesterol conjugated siRNA were packaged using the Lipofectamine 2000 (Invitrogen), tail vein injected at a dose of 2.5 mg/kg in ~0.3 mL PBS. The sense siRNA strand used for Qa-1 was: GAAGAGGAGGAGACA-CAUA. SiRNAs were synthesized by GenePharma (Shanghai, China). Primers for Qa-1 mRNA were: H2-T23-F: CCTCCATCCACTGTCTCCAA, H2-T23-R: ACCTA TGTGTCTCCTCCTCTTC; GAPDH mRNA: Gapdh-F: ACGGCCGC ATCTT CTTGTTGCA, Gapdh-R: ACGGCCAAATCCGTTCACACC.

**Murine hepatic non-parenchymal cell preparation**. NPCs were isolated as previously described with minor modifications[68]. In brief, liver tissues were passed through a 200-gauge stainless steel mesh in Hank's balanced salt solution (GIBCO). The cell suspension was centrifuged at 500 g for 5 min, and the resulting cell pellets were re-suspended in 5 mL 35% Percoll (GE Life Science), centrifuged at $500 \times g$ for 10 min at room temperature. The cell pellets containing leukocytes were re-suspended in 1 mL red blood cell lysis solution (BD) on ice for 2 min. After washed twice in RPMI 1640 containing 5% FBS, the yielded NPCs were tested for viability (more than 95%) by trypan blue exclusion.

**Peptides and IFN-γ ELISpot assay**. The MHC class I restricted (H-2-K$^b$, H-2-K$^d$, and H-2-K$^k$) peptides from HCV Core, E2, NS3, NS4, NS5A, and NS5B antigens (8–11 amino acids, overlapping peptides) were predicted using IEDB (http://www.iedb.org). Based on the calculated binding affinity to MHC, epitopes were selected for ELISpot assays (IC$_{50}$ < 50 nM, Supplementary Table 1). Where patient sera positive for HCV1b were used for infection, an array of 376 overlapping peptides (individual length of 18 amino acids, with 10 amino acids overlaps), covering the entire HCV polyprotein was used[69]. In the ELISpot assay 20–30 peptides were pooled to cover NS3, NS4B, NS5A, and NS5B, respectively. (for pool information see Supplementary Table 2). All peptides were synthesized (GL Biochem, Shanghai) and reconstituted in DMSO as stocks.

Anti-IFN-γ Enzyme-linked ImmunoSpot (ELISpot) assays were performed according exactly to manufacturer's protocol (BD). Specifically, splenocytes (4 × 10$^5$ in RPMI supplemented with 10% FBS, 50U/mL penicillin/streptomycin, 50 mM β-mercaptoethanol) were seeded on IFN-γ antibody coated 96-well PVDF membrane plates (IFN-γ ELISpot Set, BD). Individual HCV peptide (final concentration 4 μg/mL) was then added to corresponding wells. PHA and OVA peptides were used as positive or negative control, respectively. After incubation for 2.5 d, the IFN-γ secreting spots were developed and were counted in a blind fashion outsourced to Dakewe (Shenzhen, China), using an automated ELISpot reader system (Cellular Technology). Data were analyzed with software Immunospot 5.0.32. Sensitivity >50 spots/4 × 10$^5$ cells was considered as positive in IFN-γ production and IFN-γ secreting spots of OVA peptides were subtracted as background.

**Primary hepatocytes infection**. PHT were isolated from C/O$^{Tg}$ mice by a two-step collagenase perfusion protocol. In brief, the hepatic portal vein was ligated and perfused first with HBSS containing 5 mM EDTA without Ca2$^+$ and Mg2$^+$ (Beyotime Co., China), then with HBSS containing 0.025% type IV collagenase (Sigma-Aldrich). After perfusion, the liver was excised and hepatocytes were suspended in serum-free DMEM (Gibco, MD, USA) and passed through a 100-μm strainer. The filtrate was centrifuged ($50 \times g$ at 4 °C for 3 min), and the pelleted hepatocytes were re-suspended in WEM containing 10% FBS (Gibco, MD, USA). PHTs were seeded (2 × 10$^5$/mL) in 24 well plates and rested overnight, then infected with HCV J399EM (MOI = 1) as described previously[19]. Three days later, fresh isolated NPCs or purified NK cells were added and co-culture for 6 h. NK cell function was evaluated by Yac-1 cell or PMA/ ionomycin stimulation assay. To obtain HCV-conditioned medium, the supernatants were collected 72 h post HCV infection.

**Flow cytometry**. Hepatic NPCs were isolated at indicated time post HCV infection as previously described[19]. Cells suspended in PBS containing 2% FBS (Gibco) were counted using Beckman Z2 counter (Beckman). For surface staining, cells were incubated with Fc blocker (1:100, clone 93, Biolegend) and indicated antibodies in FACS buffer at 4 °C for 40 min. To measure intracellular proteins, cells were first

stained with the indicated surface antibodies, then fixed and permeabilized in cytofix/cytoperm (eBioscience) on ice for 30 min. After washed with perm/wash buffer (eBioscience), cells were incubated with antibodies towards indicated intra-cellular antigens at 4 °C for 1 h. Multi-color flow cytometry was performed on LSRFortessa (BD Biosciences, NJ). The following antibodies were from BioLegend and used according to the manufacturer's instructions: FITC-CD4 (GK1.5, 1:400), FITC-NK1.1 (PK136, 1:400), BV605-CD8α (53-6.7, 1:400), PECy7-CD3ε (145-2C11, 1:400), PECy7-F4/80 (BM8, 1:400), APCCy7-CD11b (M1/70, 1:400), PerCPCy5.5-DX5 (HMα2, 1:400), PECy7-KLRG1 (2F1/KLRG, 1:200), PECy7-CD27 (LG.3A10, 1:400), APCCy7-CD107a (1D4B, 1:200), Bv421-NKp46 (29A1.4, 1:400), APC-Ly49D (4E5, 1:400), Bv605-CD69 (HE1.2F3, 1:200), APC-Granzyme B (GB11, 1:200), PB-Ly49H (3D10, 1:400), PB-Ly49A (YE1/48.10.6, 1:400), PECy7-Ly49C/F/I/H (14B11, 1:400), APCCy7-Ly49G (AT8, 1:400), PE-Tim3 (B8.2C12, 1:200), APC-PD-1 (29F.1A12, 1:200), PECy7-TIGIT (1G9, 1:200). The following antibodies were from eBioscience and used according to the manufacturer's instructions: APC-NKG2D (CX5, 1:200), Biotin-NKG2A (20D5, 1:200), Biotin-Qa1 (6A8.6F10.1A6, 1:200), PE-IFN-γ (XMG1.2, 1:200). For intrahepatic IFN-γ expression, NPCs (1 × $10^6$) isolated from C/O$^{Tg}$ liver, were incubated with 50 ng/mL PMA and 1 μg/mL ionomycin in 200 μL RPMI complete medium for 6 h. BFA (eBioscience) was added (1:1000) 1 h post co-culture. Surface and intracellular protein expressions were examined by FACS after staining. Data of at least three independent experiments were analyzed with FlowJo Software (Tree Star, version 7.6, OR).

**NK cell-mediated cytotoxicity assay.** NPCs (1 × $10^6$) or MACS purified NK cells (2 × $10^5$) were mixed with Yac-1 cells (1 × $10^5$) or stimulated with 50 ng PMA and 1 μM ionomycin in 200 μL RPMI medium for 6 h. BFA (eBioscience) was added (1:1000) 1 h post co-culture to block cytokine secretion and meanwhile, BV421-conjugated anti-CD107a antibody was added at the beginning of incubation to detect lysosome synthesis. After stimulation, cells were harvested for detection of intra-cellular IFN-γ. First, the cells were stained for NKp46 and CD3, and then fixed/permeabilized with BD Cytofix/Cytoperm Buffer (BD Biosciences). These cells were then stained with anti-IFN-γ-PE. Data were collected on LSRFortessa (BD Biosciences, NJ) and analyzed with FlowJo Software (Tree Star, version 7.6, OR).

**Cytokine measurement.** Indicated cytokines and chemokines in serum were analyzed by Procartaplex Mouse 24-plex kit (eBioscience). In brief, sera (10 μL) from HCV-infected wt or C/O$^{Tg}$ mice were first incubated with 24-plex beads (25 μL), 10 μL detection antibody and 20 μL streptavidin-PE were then added sequentially with interval washes. Samples prepared in 120 μL sheath fluid were analyzed on a FLEXMAP 3D system with xPONENT software (Luminex).

**Statistical analysis.** Power: In compliance with ethical guidelines to minimize the number of animals used, we usually used a minimum of three mice for each data point (except indicated in figure legend) to ensure statistical power. The sample size is thus determined according to the duration of experiments and number of data points pre-determined. Batches of infection were carried out to ensure accuracy, repeatability and enough animals for each data point.

Randomization: Measures had been taken to ensure randomization. Mice are grouped with the matched age, gender, body weight, and timing of experiments, between two cohorts. To form the cohort, each batch of infection, for example, contained the number of mice more than the number of data points, to ensure the randomization and accidental exclusion of animals. The cohort was then grouped using randomly selected animals from the batches of infection. In vitro analyses (HCV copies, cytokines, and chemokines) were usually performed on specimen (sera, liver) from animals at each time point to ensure a minimal three biological replicates.

Statistics: Routinely, data collection and data analysis were performed by different persons, to blind the potential bias. All measurements data are expressed as mean ± SD to show maximal derivations, unless otherwise specified. Differences between two groups were assessed using unpaired two-tailed Student t-test. For multi-time-points data, the two-way ANOVA procedure was applied. Statistical tests are justified as appropriate. Data were in normal distribution, after variation within each group of data were estimated. The variance usually was within the range of statistic similarity. P-values <0.05 were considered significant. Statistical analyses were performed using GraphPad Prism.

**Reporting Summary.** Further information on experimental design is available in the Nature Research Reporting Summary linked to this article.

## Data availability

The authors declare that the data supporting the findings of this study are available within the paper and its Supplementary Information files, or from the authors on reasonable request. Primary data for the graphs and statistical analyses are found in the Source Data file.

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

## Acknowledgements

We thank Drs. Zhong-jun Dong (Institute for Immunology and School of Medicine, Tsinghua University) and Fu-sheng Wang (Treatment and Research Center for Infectious Diseases, Beijing 302 Hospital) for the insightful critics of the manuscript. We also thank Dr. Peng-yan Xia (Institute of Biophysics, Chinese Academy of Sciences) for help on NK killing assay and Zhi-yong Zhuo (Institute of Biophysics, Chinese Academy of Sciences) for breeding and genotyping of transgenic mice. This work was supported in part by grants from the National Natural Sciences Foundation of China to H.T. (31621061 and 81530067) and H.-r.C. (31700786); the National Basic Research Program of China to J.-q.N. (2015CB554300) and C.Z. (2015CB554303); and CAS to H.T. (QYZDJ-SSW-SMC026, 153831KYSB20160038, and XDB2903000).

## Author contributions

H.T. conceived the study. C.Z., H.-r.C. and X.-m.W. designed and performed experiments and analyzed data. S.-r.L., W.-h.W., S.-y.Z., S.-f.W. and J.-z.C. performed experiments and analyzed data. T.T., X.J., Y.-z.W., X.-w.C., S.-d.W. and J.-q.N. helped experiment designs and discussed the data. H.T., H.-r.C. and C.Z. drafted and revised the manuscript.

## Additional information

**Competing interests:** The authors declare no competing interests.

