## [Peer Review File · Nature Communications]

Reviewers' comments:

Reviewer #1 (HCV, NK)(Remarks to the Author):

Using a transgenic ICR murine model with human CD81 and Occludin specifically expressed in hepatocytes developed by Tang's Group, which has been convincingly shown to be permissive for HCV infection, Zhang et al. provide evidence of intrahepatic NK and CD8+ T cell exhaustion, the latter being irreversible, by PD-1 check-point inhibition, an established finding in this setting in humans (Gastroenterology 2012;143:1576-1585, to be cited). However, at the same time, they provide important evidence that HCV induces upregulation of Qa-1, the equivalent of HLA-E in humans and the ligand of the NKG2A inhibitory receptor, on hepatocytes, resulting in NK cell exhaustion and HCV persistence. Inhibition of the NKG2A/Qa-1 axis reversed NK cell exhaustion and NK cells were also able to restore HCV-specific CD8+ T cell responses.

The results are novel and interesting. However, several specific points require attention.

. The murine model published by the Group appears to faithfully recapitulate human disease without the need of innate immunity knock out. There is one difference though, the generally lower level of HCV replication, which does not apparently affect the ability to induce NK and CD8+ T cell exhaustion. Did the authors try to look for other exhaustion molecules on NK cells, eg Tim-3 (sought for only on CD8 T cells ?) and TIGIT, which bind Galectin-9 and CD155 or CD112, respectively ?

. It has been shown that there is a substantial difference between peripheral and intrahepatic NK cells in chronic HCV infection in humans, the former being activated by type I-IFNs and hyperexpressing TRAIL, a phenotype associated with bias toward cytotoxicity rather than cytokine secretion. I do not see any PB NK study which should corroborate this small animal model as representative of HCV infection in humans.

. KIR genes are major immune regulators of NK biology in humans. Did the authors study Ly49 family molecules in their model ?Indeed, despite the major structural differences between KIR and Ly49 proteins, the central properties of the two gene families are remarkably similar. I wonder whether this has been considered at all in this study.

. Impairment of NK cell function by HCV: cell-associated vs cell-free virus. The requirement of cell-associated HCV to induce NK cell functional changes has been extensively shown previously. However, a study carried out in humans clearly showed that there was increased TRAIL and CD69 expression following exposure of PB NK cells to virus-infected cells but not to cell-free virus in patients with chronic HCV infection (J Hepatol 2017;66:1130–1137). This emphasizes the need to examine differences between the liver and peripheral NK populations.

. It will be important to show that siRNA knock-down of Qa-1 results in rescue of both NK cell and CD8T cell functions.

. M&M, p. 33. The description of the cytotoxicity assay is unclear. Was this a degranulation (CD107a expression) or lytic assay and if so which dye was used ? The description only reports a classical ICS assay.

Reviewer #2 (T exhaustion, virus infection, immune suppression)(Remarks to the Author):

This is an interesting study exploiting a new mouse model for HCV based on immunocompetent

transgenic mice which can (like humans) become chronically infected or clear the virus shortly after infection. The aim is to explore immunotherapy in the mouse model based on blockade of inhibitory receptors on T cells and NK cells. The authors identify blockade of the NK inhibitory receptor NKG2A as a potent immunomodulator in this system, with knock-on impacts on CD8+ T cells and viral control.

It is clearly a valuable and unique system to explore the immunology of HCV and chronic hepatitis. The findings are of immunological interest although as far as immunotherapy goes it is quite hard to imagine it is likely to have so much impact in a setting where antivirals are so effective. Overall the basic principle of NKG2A/Qa1 inhibition enhancing resolution of viral hepatitis has been addressed before in a murine chronic hepatitis model of HBV (reference 41).

1. I think the authors could do more to explore what is happening in well-established chronic infection (e.g. at least a month).. There are interesting data from treatment instigated at 2 weeks, but this is also quite an acute timepoint after challenge. If the effect goes away over time that does not really seem like a problem for the hypothesis, and if it stays that would be really interesting and relevant. It would definitely be worth knowing as increasing the overall antiviral response during acute infection and improving outcome seems logical – but fixing an established problem by blockade of a single molecule would have more impact (also more relevant to their later discussion).
2. It is quite hard to fully understand what is happening to the CD8 response in this model without any tetramer staining. It does not have to be done for every epitope but one or two tracked with phenotypes including both PD1 and also NKG2A (c/e) would clarify how it is the Elispot responses appear following or accompanying effective containment of the virus.
3. Figure 4 looks at the mechanisms involved. The authors set up a primary hepatocyte model – it would be good to have a bit more information about the viral growth and % infection in this model as it is generally so hard to do. The data in Fig 4C are quite processed and it would be a bit easier to understand if they showed the actual functional data (also the spread and individual data points here - and indeed throughout - would be very useful). In these experiments it probably would be easiest to interpret the model if NK cells were used rather than the mixed populations of which NK cells are only a fraction (these are only done later, but in an experiment with no real phenotype).
4. Most important for this model is that the hepatocytes signal via Qa1 to NK cells and that this stops them responding to the targets. Given there are other cells in there it could be from other APCs – so the purified cell experiment above would be helpful, but additionally some other data in particular staining of the hepatocytes for Qa1 would be valuable to be really sure it is there and substantially upregulated at a protein level. There is some staining indicated in Figure 4F but it would be helpful to show the raw FACS plots for the hepatocytes (if possible an histochemical stain too) and a regular Qa1+ target (other papers have looked at eg B cells and CD11c+ cells). Also these data are from in vivo experiments so some data on the levels in the in vitro model (i.e as in Fig 4A) would be most relevant.
5. It seems maybe unexpected that expression of Qa1 on one cell would limit killing of/activation by another Qa1 negative cell in the in vitro model. The authors describe this as exhaustion, but that doesn't really seem like the right explanation or description of something happening in vitro over 3 days. A simple experiment using only the hepatocytes and the NK cells co-cultured to observe what phenotypic changes are seen would possibly help describe their status. Some of this is shown in Fig 4B but this doesn't seem to indicate any activation of the NK cells with the hepatocytes only, which would be part of an exhaustion process. Function could also be tested in vitro in such an experiment without the Yac cells using other stimuli. Are there any data to indicate how long NKG2A triggering in vitro impacts on the function of an NK cell? This 2-cell type experiment could also be used to track the impact of NK cells on infection of HCV in vitro directly which would really strengthen the conclusion about mechanism.
6. The primary mechanism leading to recovery of CD8s is described as via IFN γ (Fig 5). This bit is a

little hard to interpret as presumably IFNg from any source (NK or CD8 or CD4) has an antiviral effect and the greater the viral load the lower the functional capacity of the CD8 T cells. The authors don't really make it clear how they think this is happening but perhaps they can clarify this in the text and ideally experimentally and additionally show the control of anti-IFNg alone in vivo (Fig 5B).

7. Generally for human HCV studies the changes in VL that are significant for outcomes are shown on log scales – I think this would be helpful here too unless there is a good reason why not.

8. Some further virologic data on the infections using patient isolates would be helpful to understand this completely. It was actually a bit confusing to have the data from the patient 1b isolates mixed in with the other data in Figure 1 and it would I think be easier to just do one first and then the other both in the text and in the figures

Reviewer #1 (HCV, NK)(Remarks to the Author):

Using a transgenic ICR murine model with human CD81 and Occludin specifically expressed in hepatocytes developed by Tang's Group, which has been convincingly shown to be permissive for HCV infection, Zhang et al. provide evidence of intrahepatic NK and CD8+ T cell exhaustion, the latter being irreversible, by PD-1 check-point inhibition, an established finding in this setting in humans (Gastroenterology 2012;143:1576-1585, to be cited). However, at the same time, they provide important evidence that HCV induces upregulation of Qa-1, the equivalent of HLA-E in humans and the ligand of the NKG2A inhibitory receptor, on hepatocytes, resulting in NK cell exhaustion and HCV persistence. Inhibition of the NKG2A/Qa-1 axis reversed NK cell exhaustion and NK cells were also able to restore HCV-specific CD8+ T cell responses.

We thank the referee for his marks on the critical and novel finding of NKG2a/Qa1 axis in NK exhaustion. We have also included the aforementioned paper in the revised MS (ref 9).

The results are novel and interesting. However, several specific points require attention.

1. The murine model published by the Group appears to faithfully recapitulate human disease without the need of innate immunity knock out. There is one difference though, the generally lower level of HCV replication, which does not apparently affect the ability to induce NK and CD8+ T cell exhaustion. Did the authors try to look for other exhaustion molecules on NK cells, eg Tim-3 (sought for only on CD8 T cells?) and TIGIT, which bind Galectin-9 and CD155 or CD112, respectively ?

The viral loads are relatively low in serum but very high in the liver. As suggested, we showed in the revised **Figure 2D** that TIGIT expression was also up-regulated on NK cells upon HCV infection. Other inhibitory receptors, e.g., Ly49s and Tim-3, remained unaltered before or after HCV infection (**Figure S4**).

2. It has been shown that there is a substantial difference between peripheral and intrahepatic NK cells in chronic HCV infection in humans, the former being activated by type I-IFNs and hyperexpressing TRAIL, a phenotype associated with bias toward cytotoxicity rather than cytokine secretion. I do not see any PB NK study which should corroborate this small animal model as representative of HCV infection in humans.

In the revised MS, we added the dynamical changes of NKG2A and KLRG1 expression on peripheral NK cells after HCV infection, which were in line with their tendency on intrahepatic NK cells (revised **Figure 2E**), albeit at a relatively lower levels. TRAIL expression, however, remained steady and low (< 1%) in peripheral NK cells (in the revised **Figure S5A**), more or less representing the overall impaired function of peripheral NK cells. There is a vigorous but very transient activation of I-IFNs upon HCV infection in C/O^{Tg} mice (**ref 21**).

3. KIR genes are major immune regulators of NK biology in humans. Did the authors study Ly49 family molecules in their model? Indeed, despite the major structural differences between KIR

and Ly49 proteins, the central properties of the two gene families are remarkably similar. I wonder whether this has been considered at all in this study.

Per suggestion, we monitored Ly49 family, and have included Ly49D/H results in the revision (Figure 2C). Other inhibitory Ly49 family molecules (Ly49 A/C/F/G/I) are shown in Figure S4. Overall, Ly49 family proteins were not responsive to HCV infection.

4. Impairment of NK cell function by HCV: cell-associated vs cell-free virus. The requirement of cell-associated HCV to induce NK cell functional changes has been extensively shown previously. However, a study carried out in humans clearly showed that there was increased TRAIL and CD69 expression following exposure of PB NK cells to virus-infected cells but not to cell-free virus in patients with chronic HCV infection (J Hepatol 2017;66:1130–1137). This emphasizes the need to examine differences between the liver and peripheral NK populations.

We agree with the referee that HCV infected hepatocytes, rather than HCV particles *per se*, are important to induce NK exhaustion. We emphasized this point in the revised Discussion (page 17 Lines 1-6). The aforementioned literature has also been cited in the revision (ref 35).

5. It will be important to show that siRNA knock-down of Qa-1 results in rescue of both NK cell and CD8T cell functions.

Per suggestion, we've used siRNA to knock-down of Qa-1 in hepatocytes, and the results showed that down-regulation of Qa-1 in hepatocytes resulted in an effective rescue of NK function and clearance of HCV (revised Figure 6)

6. M&M, p. 33. The description of the cytotoxicity assay is unclear. Was this a degranulation (CD107a expression) or lytic assay and if so which dye was used? The description only reports a classical ICS assay.

We've corrected the description of NK cytotoxicity assays in the revised M&M section. In brief, Yac-1 cells or PMA/ionomycin were used to stimulate NK cells. NK cytotoxicity was then evaluated by FACS on cell surface CD107a and intracellular IFN- γ , respectively.

Reviewer #2 (T exhaustion, virus infection, immune suppression)(Remarks to the Author):

This is an interesting study exploiting a new mouse model for HCV based on immunocompetent transgenic mice which can (like humans) become chronically infected or clear the virus shortly after infection. The aim is to explore immunotherapy in the mouse model based on blockade of inhibitory receptors on T cells and NK cells. The authors identify blockade of the NK inhibitory receptor NKG2A as a potent immunomodulator in this system, with knock-on impacts on CD8+ T cells and viral control.

It is clearly a valuable and unique system to explore the immunology of HCV and chronic hepatitis. The findings are of immunological interest although as far as immunotherapy goes it is quite hard

to imagine it is likely to have so much impact in a setting where antivirals are so effective. Overall the basic principle of NKG2A/Qa1 inhibition enhancing resolution of viral hepatitis has been addressed before in a murine chronic hepatitis model of HBV (reference 41).

DAA's are so effective that the medical/industrial sectors become optimistic for "Hep C cure" nowadays. However, the related fields have also raised concerns of several blocks of DAA's (such as drug resistance, progression of cirrhosis/HCC, unwanted HBV activation in HBV/HCV co-infection, *etc.*), that would call for a combination of immunotherapies with DAA's in the future. In terms of the novelty of NKG2A/Qa1, our paper differs in two substantial aspects than the HBV study in the mentioned reference (ref 40 in the revised MS). (1), we are using a true HCV infection model, instead of a HBV genome amplification model (there is no evidence that HBV virions amplified by AAV carrier can infect the liver). Thus we found that NKG2A/Qa-1 checkpoint is essential for the establishment of HCV persistent infection in the liver. In contrast, the conclusion that ref 41 can make is that NKG2A/Qa-1 interaction may associate with AAV/HBV genome replication in the liver. (2), more critically, we provided convincing evidence that NKG2A/Qa-1 is a NK checkpoint that accounts for NK and CD8 T exhaustion, and NKG2A/Qa-1 inhibition revigorate both NK and CD8 T cytotoxicities toward HCV clearance. The aforementioned studies on HBV came in short of the mechanism.

1. I think the authors could do more to explore what is happening in well-established chronic infection (e.g. at least a month). There are interesting data from treatment instigated at 2 weeks, but this is also quite an acute timepoint after challenge. If the effect goes away over time that does not really seem like a problem for the hypothesis, and if it stays that would be really interesting and relevant. It would definitely be worth knowing as increasing the overall antiviral response during acute infection and improving outcome seems logical – but fixing an established problem by blockade of a single molecule would have more impact (also more relevant to their later discussion).

We chose 2 weeks after HCV infection to start the anti-NKG2A treatment based on two observations in this mice model: (1) without any intervention (DAA or anti-NKG2A), HCV infection will persist for more than 16 months or more; (2) 2 *wpi* is the turning point when HCV persistent infection starts (ref. 21), and at this point, NKG2A upregulation in NK cells and PD-1/Tim-3 upregulation in T cells prevailed. Therefore, the revised paper would like to focus on the role of Qa-1/NKG2A checkpoint in the establishment of HCV persistence infection. To more precisely reflect the mechanism, we've also revised the title to "Tuning of NK cell function by NKG2A leads to CD8⁺ T cell exhaustion and HCV persistence".

We leave the therapeutic effect of anti-NKG2A to another ongoing project, which studies whether anti-NKG2A treatment at a much latter phase of chronic infection (1 *mpi* and 4 *mpi*), alone or in combination with DAA's, would clear HCV and protect the host from re-infection by boosting memory immune responses. Before we publish these results in a different paper, we can only ensure the referee at this stage that anti-NKG2a works well even at 1 (steatosis) or 4 months (fibrosis) after HCV infection.

2. It is quite hard to fully understand what is happening to the CD8 response in this model without any tetramer staining. It does not have to be done for every epitope but one or two tracked with phenotypes including both PD1 and also NKG2A (c/e) would clarify how it is the ELISpot responses appear following or accompanying effective containment of the virus.

We agree with the reviewer and we'd tried very hard but in vain the tetramer staining. That's because, unlike the inbred C57/BL6 (H2-Kb) or Balb/c (H2-Kd) mice, ICR is an outbred strain with mixed H2 haplotypes (H2-Kb, H2-Kq, H2-Kd *etc.*) in different combinations, and their expression levels vary among litters. Therefore, tetramer results basically cannot reproduce from batch to batch due this H2 haplotype heterogeneity in ICR strains. Before we can finally establish a colony of engineered ICR strain with a fixed and stable H2 haplotype, ELISpot assays with various epitopes may serve to maximally reflect the overall function of HCV multi-specific T cells.

3. Figure 4 looks at the mechanisms involved. The authors set up a primary hepatocyte model – it would be good to have a bit more information about the viral growth and % infection in this model as it is generally so hard to do. The data in Fig 4C are quite processed and it would be a bit easier to understand if they showed the actual functional data (also the spread and individual data points here - and indeed throughout - would be very useful). In these experiments it probably would be easiest to interpret the model if NK cells were used rather than the mixed populations of which NK cells are only a fraction (these are only done later, but in an experiment with no real phenotype).

We routinely isolate and culture mouse primary hepatocytes for HCV infection *in vitro*, with satisfactory/reproducible viral growth and viral infectivity, as described in our previous work (ref 21). Per suggestion, we added viral infection and infectivity of virus particles in the revised Figure S8. We've also revised Figure 5C and Figure 5E with easier layouts for better understanding. Also in Figure 5E, we showed that HCV-infected PHT efficiently inhibited NK cells but not NPCs purified from the liver.

4. Most important for this model is that the hepatocytes signal via Qa1 to NK cells and that this stops them responding to the targets. Given there are other cells in there it could be from other APCs – so the purified cell experiment above would be helpful, but additionally some other data in particular staining of the hepatocytes for Qa1 would be valuable to be really sure it is there and substantially upregulated at a protein level. There is some staining indicated in Figure 4F but it would be helpful to show the raw FACS plots for the hepatocytes (if possible an histochemical stain too) and a regular Qa1+ target (other papers have looked at eg B cells and CD11c+ cells). Also these data are from *in vivo* experiments so some data on the levels in the *in vitro* model (i.e. as in Fig 4A) would be most relevant.

We've followed carefully the suggestion and provided necessary data in the revision. First, we measured Qa-1 protein expression in hepatocytes both *in vitro* (Figure S8C) and *in vivo* (Figure 6A), which showed that HCV infection upregulated Qa1 expression in hepatocytes, either purified or *in situ* of livers. Second, FACS analysis showed that HCV-increased Qa-1 expression is restricted to hepatocytes but not B, T, DC or CD11b⁺ myeloid cells in liver (Figure 6B).

5. It seems maybe unexpected that expression of Qa1 on one cell would limit killing of/activation by another Qa1 negative cell in the *in vitro* model. The authors describe this as exhaustion, but that doesn't really seem like the right explanation or description of something happening *in vitro* over 3 days. A simple experiment using only the hepatocytes and the NK cells co-cultured to observe what phenotypic changes are seen would possibly help describe their status. Some of this is shown in Fig 4B but this doesn't seem to indicate any activation of the NK cells with the hepatocytes only, which would be part of an exhaustion process. Function could also be tested *in vitro* in such an experiment without the Yac cells using other stimuli. Are there any data to indicate how long NKG2A triggering *in vitro* impacts on the function of an NK cell? This 2-cell type experiment could also be used to track the impact of NK cells on infection of HCV *in vitro* directly which would really strengthen the conclusion about mechanism.

To avoid misunderstanding, we changed the description of NK "exhaustion" to "inhibition" to more precisely interpret these *in vitro* assays. Per suggestion, we've also added another NK function assay using PMA plus ionomycin instead of Yac1 cells for NK cell stimulation, which led to the same conclusion (**Figures 5B & 5D**). In such *in vitro* co-culture system, unfortunately, NK cells seemed not able to affect HCV replication or kill those HCV infected hepatocytes.

6. The primary mechanism leading to recovery of CD8s is described as via IFN γ (Fig 5). This bit is a little hard to interpret as presumably IFN γ from any source (NK or CD8 or CD4) has an antiviral effect and the greater the viral load the lower the functional capacity of the CD8 T cells. The authors don't really make it clear how they think this is happening but perhaps they can clarify this in the text and ideally experimentally and additionally show the control of anti-IFN γ alone *in vivo* (Fig 5B).

As for the cell source of IFN- γ , we've provided more data in the revision that only NK, NKT and T cells were IFN- γ -positive cells in the liver, but importantly, IFN- γ was increased only in NK cells after anti-NKG2A treatment (**Figure 7A**). Therefore, IFN- γ would primarily come from reactivated NK cells after anti-NKG2A.

It is true that a reduction of viral burden can help recover the functional capacity of CD8 T cells. However, we showed that this recovery process required IFN- γ , because CD8 T cells remained inactive after IFN- γ neutralization, even NK cells had been rejuvenated and viral load eventually reduced by anti-NKG2A treatment. Of course, we do not want rule out the contribution of the reduced viral load that work together with increased IFN- γ to activate CD8 T cells.

7. Generally for human HCV studies the changes in VL that are significant for outcomes are shown on log scales – I think this would be helpful here too unless there is a good reason why not.

We have made modifications as suggested in the revised MS.

8. Some further virologic data on the infections using patient isolates would be helpful to understand this completely. It was actually a bit confusing to have the data from the patient 1b

isolates mixed in with the other data in Figure 1 and it would I think be easier to just do one first and then the other both in the text and in the figures

We have moved the data of patient HCV1b to the revised **Figure S2**.

In summary, we have done our best to extensively revise the paper by performing many experiments, especially the difficult *in vivo* Qa-1 knockdown experiment. We believe the revision has addressed all the concerns by the referees and clarified all the confusing descriptions and layouts in the paper.

We thank you for your confidence in our work and your patience in the revision process.

Best regards,

Hong

REVIEWERS' COMMENTS:

Reviewer #1 (Remarks to the Author):

The authors have satisfactorily answered my concerns. No further comments.

Reviewer #2 (Remarks to the Author):

The authors have addressed all of the points raised. Figure 6B could have an x axis to describe the stain on the figure.

REVIEWERS' COMMENTS:

Reviewer #1 (Remarks to the Author):

The authors have satisfactorily answered my concerns. No further comments.

AU: Thank you for your kindly advises throughout the review process.

Reviewer #2 (Remarks to the Author):

The authors have addressed all of the points raised. Figure 6B could have an x axis to describe the stain on the figure.

AU: Thank you for your kindly advises throughout the review process. We have added an x axis for Figure 6B as well as Supplementary Figure 9B accordingly.